# Flexible Resistive Gas Sensor Based on Molybdenum Disulfide-Modified Polypyrrole for Trace NO_2_ Detection

**DOI:** 10.3390/polym16131940

**Published:** 2024-07-07

**Authors:** Kuo Zhao, Yunbo Shi, Mingrui Cui, Bolun Tang, Canda Zheng, Qinglong Chen, Yuhan Hu

**Affiliations:** 1School of Measurement and Control Technology and Communication Engineering, Harbin University of Science and Technology, Harbin 150080, China; iridescen7@163.com (K.Z.); cuimingrui2022163@163.com (M.C.); tangbolun@sina.com (B.T.); zhengcanda@outlook.com (C.Z.); 16601373789@163.com (Q.C.); 2China Higher Educational Key Laboratory for Measuring & Control Technology and Instrumentations of Heilongjiang Province, Harbin 150080, China; 3The 49th Research Institute of China Electronics Technology Group, Harbin 150001, China; huyuhan0619@163.com

**Keywords:** flexible sensor, polypyrrole, fluorene polyester, molybdenum disulfide, NO_2_ sensor

## Abstract

High sensitivity and selectivity and short response and recovery times are important for practical conductive polymer gas sensors. However, poor stability, poor selectivity, and long response times significantly limit the applicability of single-phase conducting polymers, such as polypyrrole (PPy). In this study, PPy/MoS_2_ composite films were prepared via chemical polymerization and mechanical blending, and flexible thin-film resistive NO_2_ sensors consisting of copper heating, fluorene polyester insulating, and PPy/MoS_2_ sensing layers with a silver fork finger electrode were fabricated on a flexible polyimide substrate using a flexible electronic printer. The PPy/MoS_2_ composite films were characterized using X-ray diffraction, Fourier-transform infrared spectroscopy, and field-emission scanning electron microscopy. A home-built gas sensing test platform was built to determine the resistance changes in the composite thin-film sensor with temperature and gas concentration. The PPy/MoS_2_ sensor exhibited better sensitivity, selectivity, and stability than a pure PPy sensor. Its response to 50 ppm NO_2_ was 38% at 150 °C, i.e., 26% higher than that of the pure PPy sensor, and its selectivity and stability were also higher. The greater sensitivity was attributed to p–n heterojunction formation after MoS_2_ doping and more gas adsorption sites. Thus, PPy/MoS_2_ composite film sensors have good application prospects.

## 1. Introduction

With the rapid development of modern industry, environmental pollution has increased significantly, and large volumes of toxic and harmful gasses are continuously released into the environment that sustains our existence, affecting the air quality and causing various problems [1,2,3,4,5]. To prevent these gaseous emissions from harming humans and the environment, toxic gas monitoring is becoming increasingly important. Among the toxic gasses that require monitoring, nitrogen dioxide (NO_2_) is a common toxic air pollutant. The main sources of NO_2_ are the combustion of certain fuels, such as coal and petroleum, biomass combustion triggered by the extreme heat of lightning during thunderstorms, and the microbial fixation of nitrogen in agriculture [6,7,8,9]. The major effects of NO_2_ include inflammation of the respiratory tract, decreased lung function, increased reactivity to allergens, and the formation of fine particulate matter (PM) and ground-level ozone—which have adverse health effects—as well as the formation of acid rain, which damages vegetation and results in the acidification of buildings, lakes, and streams [10,11,12]. The lethal concentration of NO_2_ for 50% of people who are exposed (LC50) for 1 h of exposure is approximately 174 ppm; therefore, it is important to develop sensors for the detection of nitrogen dioxide with low detection limits and high sensitivity [13,14].

In recent years, conductive polymers have frequently been used as sensing materials because of their excellent physicochemical properties. They are versatile, inexpensive, and easy to synthesize and have large surface areas, low operating temperatures, and high sensitivities. In addition, exposure to various gasses leads to changes in the electrical and optical properties of these sensing materials in addition to other properties [15]. Among the various types of conductive polymers, polypyrrole (PPy) has become a major focus of recent research by virtue of its chemical stability under atmospheric conditions, its ease of adhering to various substrates, and ease of synthesis by electrochemical and chemical methods. However, compared to inorganic semiconductor sensors, sensors made with PPy are less sensitive to target gasses and have longer response times, and these drawbacks limit the applicability of this material [16]. Therefore, it is usually necessary to optimize the properties of PPy by doping [17]. In recent years, research has shown that the mutual doping of conducting polymers and semiconducting metal oxides [18] can produce synergistic effects, with the enhancement of the performance of both materials owing to interfacial electron transfer and catalytic effects, among others. Zhang et al. prepared PPy-coated SnO_2_ hollow sphere hybrid materials via in situ polymerization using SnO_2_ hollow spheres as carriers, and the performance of the resulting composites was shown to be much better than that of pure PPy. The hybrid material exhibited a faster response and higher sensitivity to ammonia at room temperature than pure PPy [19]. Similarly, Mane et al. fabricated PPy/WO_3_ hybrid nanomaterials on glass substrates for chemoresistive NO_2_ sensors using the drop-casting method [20]. For the p-type semiconductor PPy, doping with n-type materials such as WO_3_ can generate more p–n heterostructures, and therefore, the sensor’s performance is improved.

Compared to conventional metal oxide materials, metal sulfides have unique two-dimensional (2D) atomic structures, which offer significantly increased areas of contact between the gas molecules and material surface, resulting in a greater number of active sites [21,22]. In addition, metal sulfides have higher charge mobilities, which allows carriers to migrate more easily inside the material and effectively extends the hole and electron lifetimes [23]. Among the many metal sulfides available, MoS_2_ is a typical 2D transition metal sulfide with a graphene-like layered structure that has been widely investigated as a promising candidate for various applications owing to its excellent properties [24]. Han et al. constructed a NO_2_ sensor with a unique 2D/0D MoS_2_/SnO_2_ p–n heterostructure based on MoS_2_ nanosheets on a Si/SiO_2_ substrate, which exhibited a high response value of 18.7 at a low NO_2_ concentration (5 ppm) [25]. Similarly, Liu et al. designed and fabricated a composite material based on MoS_2_ nanomaterials modified with tin disulfide (SnS_2_) by mechanical exfoliation combined with the hydrothermal method; the mechanically exfoliated MoS_2_ nanosheets acted as the central sensing channel, interacting with gaseous NO_2_, while the SnS_2_ nanoparticles decorating the nanosheet surfaces acted as a surface passivation layer to prevent the oxidation of the MoS_2_ nanosheets. Owing to the synergistic effect of the excellent gas sensing properties of the MoS_2_ nanosheets and the surface-protecting properties of the SnS_2_ nanoparticles, the prepared MoS_2_/SnS_2_ composites exhibited reliable long-term stability with a response/recovery rate of 28s/3s for the sensing of NO_2_ [26]. Ding et al. synthesized MoS_2_/rGO/Cu_2_O ternary composites on an alumina ceramic substrate using the soft stenciling method, and gas sensing tests showed a good response (14.88%) to NO_2_ for the MoS_2_/rGO/Cu_2_O composite sensor at the ppb level at room temperature. This good response is attributed to the positive synergistic effect of the three-phase composite material, as the gas permeation and diffusion channels are tightly loaded on MoS_2_/rGO nanosheets, giving the composite material a larger surface area [27].

However, in recent years, research on NO_2_ sensors has mostly been focused on the use of glass, SiO_2_, and other rigid substrate materials. Such sensors have poor mechanical flexibility, are easily damaged by mechanical impact, and have high coefficients of thermal expansion and performances that are more strongly dependent on temperature [28]; in addition, they cannot be tightly fitted onto uneven or complex surfaces. In contrast, the covalent bonding forces are very strong within metal sulfide layers, so these materials have good mechanical properties. Therefore, flexible and wearable sensors based on PPy/MoS_2_ composites have excellent potential application prospects and should therefore be developed.

In this work, we synthesized a new PPy/MoS_2_ binary composite material by chemical polymerization and mechanical blending using a flexible electronic printer, dispensing printed copper micro-heaters with silver fork finger electrodes. PPy/MoS_2_ composites were characterized using scanning electron microscopy (SEM), X-ray diffraction (XRD), and Fourier-transform infrared spectroscopy (FTIR), and the structural morphology of the composites was studied in detail. A flexible thin-film NO_2_ sensor was then prepared by coating the material onto a silver fork finger electrode on a polyimide (PI) substrate via dispensing printing. A gas sensing test platform was built to sense gas concentrations between 1 and 100 ppm. Finally, the gas sensing mechanism was investigated, and the improved sensing performance was understood.

## 2. Materials and Methods

### 2.1. Materials

Pyrrole (C_4_H_5_N) (AR grade Macklin, Shanghai, China), ammonium persulfate ((NH_4_)_2_S_2_O_8_) (AR grade, Fuchen Reagent Factory, Tianjin, China), *N*,*N*-dimethylformamide (C_3_H_7_NO) (AR grade, Macklin), methanol (CH_3_OH) (AR grade, Aladdin Co., Ltd., Shanghai, China), silver carbon paste (Shenzhen Saiya Electronic Paste Co., Ltd., Shenzhen, China), pinoresinol (C_10_H_18_O) (AR grade, Tianjin Fuchen Reagent Factory), and molybdenum disulfide (MoS_2_) (Tianjin Zhiyuan Chemical Reagent Co., Tianjin, China) were used. Fluorene polyester (FPE) particles were purchased from PolyK (Philipsburg, PA, USA). All chemicals were used as received without further purification. 

### 2.2. Characterization Methods

The structures of PPy and PPy/MoS_2_ were analyzed using an X-ray diffractometer (Rigaku Smartlab 3kw, Tokyo, Japan) under Cu Kα radiation in the scanning range of 20–80°. The valence states of the elements were analyzed by X-ray photoelectron spectroscopy (Thermo ESCALAB 250Xi, Los Angeles, CA, USA). The morphology of the samples was observed by field-emission scanning electron microscopy (ZEISS Gemini SEM 300, Berlin, Germany) at an accelerating voltage of 3–15 kV. FTIR spectroscopy (Perkin Elmer 100, Los Angeles, CA, USA) was used for the chemical structure analysis within the frequency range of 400–4000 cm^−1^.

### 2.3. Preparation of Sensing Material Composites

First, PPy powder was prepared via chemical oxidative polymerization using pyrrole as the monomer and ammonium persulfate (APS) as the oxidant. First, pyrrole (0.7 mL) and APS powder (2.28 g) were separately dissolved in diluted hydrochloric acid (1 M, 100 mL), and then an acidic APS solution was added to an acidic pyrrole solution at the rate of 3 drops per second, and they were mixed together in a beaker for 4 h to allow the polymerization reaction to proceed. After polymerization, the precipitate obtained after vacuum filtration was washed six times with methanol and deionized water and then dried in a vacuum heating chamber at 60 °C for 12 h to obtain PPy powder [29].

PPy/MoS_2_ composites were prepared using the mechanical blending method. MoS_2_ and PPy powders were added to a smooth agate mortar in a specific ratio, ground with a pestle and mortar for 3 h, dispersed in dimethylformamide (DMF) solution (5 mL) with magnetic stirring for 3 h, and finally dried at 80 °C for 24 h in a heating chamber to obtain the PPy/MoS_2_ composite as a homogeneous mixture (Figure 1).

### 2.4. Sensor Structure Design and Gas Sensing Test Platform

The sensor was fabricated on a PI substrate using a flexible electronic printer (Scientific 3, Zhongbin Technology, Harbin, China) to perform dispensing printing. The flexible thin-film sensor substrate consisted of a carbon heating layer, an insulating layer, a silver fork finger electrode, and a sensing layer (Figure 2a,b). The insulating layer was made from FPE via the following method: FPE (0.4 g) was dissolved in *N*-methyl-2-pyrrolidone (NMP) (4 mL), and the resulting solution was stirred rigorously for 2 h before being scraped and coated onto the heating layer using a flexible printer and then dried at 80 °C for 24 h. In our previous studies [30], FPE was demonstrated to be a high-performance insulation material with a high dielectric constant (3.5) and a glass transition temperature of 335 °C, which allows it to perform well as a sensor insulation layer. The dried powder sample obtained as the PPy/MoS_2_ composite (Section 2.3) was ground to a 200 mesh size in an agate mortar and sieved. An appropriate amount of rosinol was added dropwise to the milled composite material, which was then coated onto the silver fork finger electrode layer of the sensor via dispensing printing with a flexible electronic printer. The fabricated sensor was vacuum-dried at 60 °C for 12 h and aged at its operating temperature for three days. The gas sensing test platform consisted of a gas dispenser, DC power supply, data acquisition card, and PC (Figure 2c). An amount of gas was injected into the gas chamber (1 L) using a gas dispenser to achieve a specific final concentration of the gas. A DC power supply provided the operating voltage for the carbon heater in the sensor. The adsorption of the gas onto the gas sensing material resulted in a change in the sensor resistance, and the data acquisition card recorded the change in resistance and uploaded it to the PC to record the data. The sensor film photo is shown in Appendix A.

## 3. Results and Discussion

Figure 3a shows an SEM image of the prepared PPy, and Figure 3b shows a higher-resolution image from the same field of view. The PPy sample exhibits spherical particle stacking [31] with a particle diameter of approximately 100 nm. Most of the PPy particles are uniformly distributed, and the material as a whole has porous characteristics, suggesting that gas molecules will be easily adsorbed by the sensing material. The characteristic C and O peaks are shown in Appendix A, and the weight percentages of elemental C and O were calculated to be as high as 6.55, indicating that the pyrrole monomer was fully polymerized and oxidized [32]. SEM images of the PPy/MoS_2_ samples are shown in Figure 3c–f. The structure was assembled from the PPy spherical structure and the 2D lamellar structure of MoS_2_. It can be seen that MoS_2_ is well embedded within the PPy, and a small amount of PPy is directly adhered to the MoS_2_ sheets. The distributions of C, O, Mo, and S are shown in Figure 3(f1)–(f4), which reveal that the MoS_2_ is uniformly distributed among the C and O distributions of PPy, indicating a more homogeneous distribution of PPy. In Figure 3, it can be seen that doping with the 2D MoS_2_ sheets produces many porous structures, providing additional adsorption sites in the material. According to the cross-sectional analysis in Appendix A, it can be seen that the composite sensing membrane is approximately 2 µm thick, and the introduction of MoS_2_ nanosheets adds a large number of adsorption sites to the composite material, suggesting that the gas-sensitive performance of the composite material might be superior to that of pure PPy.

Figure 4a shows the XRD patterns of the pure PPy and various PPy/MoS_2_ samples at 25 °C. It can be seen that the diffraction peaks of almost all of the binary materials are in good agreement with those of the pure PPy material, and there is a broad peak between 2*θ* = 20 and 30°, indicating that the polymer is amorphous [33]. The broad peaks were caused by the X-ray scattering of the PPy chains. The presence of these peaks in each spectrum indicates that the introduction of MoS_2_ did not disrupt the structure of PPy itself. The diffraction peaks at 2*θ* = 32.6, 39.55, 44.3, and 59° are attributed to the (100), (103), (006), and (105) planes of MoS_2_, respectively [34]. As the MoS_2_ mass fraction increased, the characteristic peaks assigned to this compound became pronounced, indicating that the MoS_2_ sheets and PPy were well mixed. Figure 4b shows the FTIR spectra of the two samples, where characteristic absorption bands are observed at 1550, 1451, 1181, 1046, 1046, and 794 cm^−1^; these were attributed to the C–C vibration in the pyrrole ring, the C–N respiratory vibrations of the pyrrole ring, and the C–N and N–H in-plane deformation vibrations, respectively [35,36]. The four composites exhibit absorption bands close to those of PPy. All of the above observed absorption properties confirm the synthesis of PPy and that its functional groups are not destroyed when it is doped with MoS_2_.

To determine a suitable working temperature for the flexible gas sensor, we measured the thermal decomposition temperatures of both PPy and PPy/MoS_2_ (10%) using a ZCT–B thermogravimetric (TG) analyzer, as shown in Figure 5. The TG and differential thermal analysis (DTA) curves were obtained at a scanning rate of 10 °C/min. The curves in Figure 5 are the TG and DTA curves, respectively. The PPy sample continuously lost weight during heating from the start (at 100 °C). The thermal decomposition of PPy can be divided into two stages: (1) the first stage of weight loss occurred at the range of 300–420 °C and resulted from the detachment of water molecules from PPy; (2) the weight loss between 420 and 650 °C was a result of the degradation of the PPy macromolecular chains. The polymer completely decomposed above 650 °C. The TG and DTA curves remained constant at temperatures above 650 °C, which indicates that the decomposition of PPy began at 300 °C [37]. Figure 5b shows the TG and DTA curves of the PPy/MoS_2_ composite. These have similar shapes to those of PPy, with two relatively symmetric and distinct heat absorption peaks at the ranges of 350–400 and 400–530 °C. The composite material lost very little weight below 250 °C. When compared with that of pure PPy, the thermal stability of PPy in the hybrid material is almost unchanged, and the TG analysis indicates that that the maximum operating temperature of the sensor can be set to 350 °C.

To investigate the effect of the MoS_2_ content on the gas sensing performance of the PPy–based NO_2_ sensors and impact resistance, five composite sensors with different MoS_2_ mass fractions (0–30%) were tested using 100 ppm NO_2_ at room temperature to observe the response. The highest response (49.8%) was obtained for the sensor made with the composite containing 10% MoS_2_, and the lowest response was obtained for that made with the 30% MoS_2_ composite. Figure 6a shows the changes in the resistance values of the different sensors after the chamber of the gas sensing test platform was charged with gas. It is apparent that as the MoS_2_ content increased, the resistance of the composite material decreased. In this study, the sensitivity (response) of the sensor was defined in terms of the resistance of the gas sensor in air *R_a_* and in the sample gas *R_g_* as follows:(1)$$\begin{array}{c} \mathrm{Response}\;\left(\%\right)=\left|\frac{R_{g}-R_{a}}{R_{a}}\right|\times 100\%. \end{array}$$

As shown in Figure 6b, the PPy/MoS_2_ sensor made with the 10% MoS_2_ composite has the highest sensitivity (49.8%) to 100 ppm NO_2_ at room temperature. However, Figure 6a shows that the response/recovery times of several of the sensors at room temperature are long, which is unfavorable in practical applications. To optimize the performance of the sensors, the 10% PPy/MoS_2_ composite was selected for subsequent experiments in which the temperature was increased in an attempt to minimize the response/recovery time of the sensor. 

As shown in Figure 7, the sensitivity of the sensor decreased as the temperature increased. By the time the temperature reached 150 °C, during a continuous increase in temperature, the sensitivity of the sensor reduced from 49.89% to 38.75%, and its response and recovery times decreased to 120 and 57 s, respectively; at room temperature, the response and recovery times were 572 and 461 s, respectively. Thus, it was demonstrated that at 150 °C, although the sensitivity of the sensor was slightly reduced, the efficiency of gas adsorption and desorption was greatly improved, and hence, the recovery time was dramatically reduced. Therefore, 150 °C can be considered a good working temperature for the sensor.

In order to investigate the performance of the sensors at low NO_2_ concentrations, the resistances of the pure PPy and 10% PPy/MoS_2_ composite sensors at 150° were measured under a concentration gradient change from 1 to 20 ppm (Figure 8). From the response curves, it can be seen that the resistance of the sensor decreased significantly as the gas concentration increased; at 1 ppm, the sensitivity of the pure PPy sensor was 2.7%, and that of the PPy/MoS_2_ sensor was 8%, while the response and recovery times of the latter were shorter. Therefore, the sensor has good resolution even at low NO_2_ gas concentrations, and it can provide a basis for the development of improved trace gas detection techniques.

As shown in Table 1, the PPy/MoS_2_ composite thin-film sensor prepared in this work has a lower operating temperature than traditional metal oxide sensors, and it has a faster response/recovery time than pure conductive polymer thin-film sensors.

Figure 9a shows the repeatability of the PPy/MoS_2_ sensor measurements for NO_2_ gas at 100 ppm, where it is apparent that the response and recovery characteristics were maintained without significant degradation. Over six cycles of measurements at 150 °C, the average response was 38%. The responses of the PPy/MoS_2_ sensor to other gasses at 50 ppm or higher concentrations at 150 °C are shown in Figure 9b. The responses of the PPy/MoS_2_ sensor to interferent gasses, such as ammonia (NH_3_), sulfur dioxide (SO_2_), hydrogen sulfide (H_2_S), and nitric oxide (NO), were lower than the response to NO_2_. These results demonstrate the excellent selectivity and repeatability of the PPy/MoS_2_ composite sensor.

The results of the long-term stability tests are shown in Figure 10. The resistance of the PPy/MoS_2_ sensor was investigated over 40 days of exposure to a fixed NO_2_ concentration of 100 ppm (measured at 5-day intervals) at the operating temperature. The PPy/MoS_2_ sensor initially showed a maximum response of 38.5%, which gradually decreased to 29% and eventually stabilized, with no further decrease in the sensitivity over time after 30 days; the overall decrease in sensitivity was approximately 25%. After 30 days, the sensitivity no longer decreased with time, and there was an overall decrease of approximately 25%. In contrast, the sensitivity of the pure PPy sensor decreased by approximately 67% after 30 days. Thus, the introduction of MoS_2_ nanosheets can improve the long-term stability of the device to some extent. Furthermore, the high reproducibility of the re-response of the PPy/MoS_2_ sensor to 100 ppm NO_2_ indicates that the MoS_2_-doped PPy gas sensor has good potential. In addition, the performance of the sensor is also affected by environmental humidity and the film bending angle, as shown in Appendix A.

PPy is a p-type semiconductor whose principal charge carriers are holes, and when the oxidizing gas NO_2_ passes over the sensor, the electrons in the polymer are trapped to form the anion NO_2_^−^, and hence, the number of holes in the material increases, which leads to a decrease in the electrical resistance. When air passes over the sensor, the absorbed NO_2_^−^ gives up its trapped electrons, and electron–hole recombination occurs. The steps of the sensing mechanism are shown in the following equations [42]:(2)$$\begin{array}{c} {NO}_{2}+e^{-}⇄{NO}_{2}^{-}, \end{array}$$
(3)$$\begin{array}{c} \mathrm{PPy}-e^{-}⇄{\mathrm{PPy}}^{+}. \end{array}$$

Doping is a key parameter that plays an important role in the gas sensing mechanisms of chemoresistive gas sensors. In the PPy/MoS_2_ hybrid nanocomposite sensor, a p-n junction formed between the p-type PPy and n-type MoS_2_. Upon exposure to oxidized NO_2_ gas (electron acceptance), the resistance of the PPy/MoS_2_ hybrid nanocomposite sensor decreased. Electron charge transfer is the main cause of the change in resistance in the PPy/MoS_2_ hybrid nanocomposite sensors upon NO_2_ gas adsorption. Upon exposure to NO_2_ gas, electron charge transfer can occur between NO_2_ and PPy because MoS_2_ nanoparticles are embedded in the PPy matrix. The interaction of NO_2_ gas molecules with the electronic network of PPy with embedded MoS_2_ nanosheets leads to a decrease in the sensor resistance [20,43,44,45]. The reduced resistance of the sensor film confirms that the PPy/MoS_2_ composite sensor behaves as a p-type semiconductor and that the charge transfer is mainly due to PPy. A schematic energy diagram of the heterojunction between PPy and MoS_2_ is illustrated in Figure 11b. PPy behaves as a p-type semiconductor with a bandgap of 2.7 eV, and MoS_2_ is an n-type semiconductor with a bandgap of 1.2 eV. When the p-type PPy and n-type MoS_2_ are in contact, the mutual diffusion of the two main carriers at the interface results in the formation of a p–n heterojunction and a self-assembled depletion layer. The adsorption and desorption of NO_2_ resulted in the modulation of the width of the depletion layer of the p–n heterojunction. When gaseous NO_2_ interacts with the surface of the PPy/MoS_2_ hybrid nanocomposite sensor, the width of the depletion layer decreases owing to the decrease in the sensor resistance, resulting in easier transport of the charge carriers in the corresponding energy band of the sensor [46,47,48]. As shown in the SEM images, the MoS_2_ nanosheets were wrapped within and interconnected with the PPy, providing electron transfer pathways during gas sensing. In addition, the introduction of 2D MoS_2_ nanosheets increased the dispersion of PPy, and the looser and more porous structure of the binary composite material had a larger surface area, which greatly facilitated gas diffusion, improving the performance of the sensor.

## 4. Conclusions

In this study, PPy/MoS_2_ binary composite materials were synthesized via chemical polymerization and mechanical blending, and their morphologies and structures were characterized using techniques including XRD, FTIR, and SEM. A flexible NO_2_ sensor was then fabricated from the composite material on a polyimide substrate using a flexible electronic printer. At room temperature, in an atmosphere of 100 ppm NO_2_, the sensitivity of the 10% PPy/MoS_2_ composite sensor was higher than that of an analogous sensor made using pure PPy (49.8% vs. 5%). However, the response and recovery times of the two sensors at room temperature were greater, and the response and recovery times of the two sensors were also greater at higher temperatures. The optimal temperature, in terms of the response and recovery times, was 150 °C, with the response time of the 10% PPy/MoS_2_ sensor being shortened from 572 s (at room temperature) to 120 s and the recovery time being shortened from 461 s (at room temperature) to 57 s. In addition, the sensor still retained relatively good resolution over low concentrations of 1–20 ppm, with a sensitivity of 8% to 1 ppm NO_2_ for the 10% PPy/MoS_2_ sensor at 150 °C. The enhanced response of the composite materials is attributed to the formation of p–n heterostructures and a self-assembled depletion layer between the p-type PPy and n-type MoS_2_. Doping with MoS_2_ resulted in a larger surface area and more contact sites between the NO_2_ gas and the material. In addition, the introduction of a heating layer resulted in a synergistic enhancement in the sensor sensitivity and response/recovery speed. Therefore, it was demonstrated that the composite material fabricated by doping PPy with MoS_2_ is an effective NO_2_ gas sensing material that provides a foundation for the future development of flexible, wearable trace gas detection devices.

## Figures and Tables

**Figure 1 polymers-16-01940-f001:**
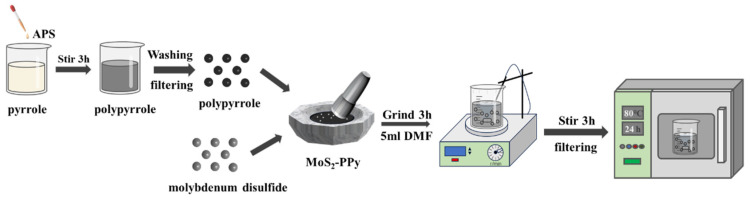
Schematic of PPy/MoS_2_ composite preparation procedure.

**Figure 2 polymers-16-01940-f002:**
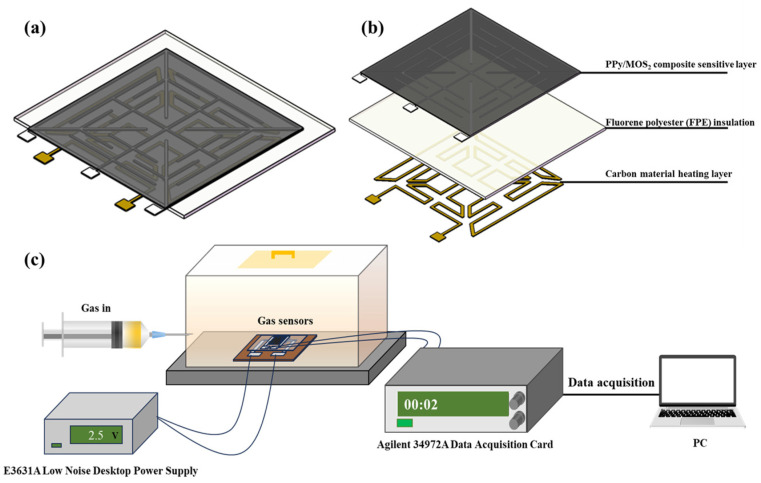
Schematics of (**a**) overall sensor structure, (**b**) multilayer sensor structure, and (**c**) gas sensing test platform.

**Figure 3 polymers-16-01940-f003:**
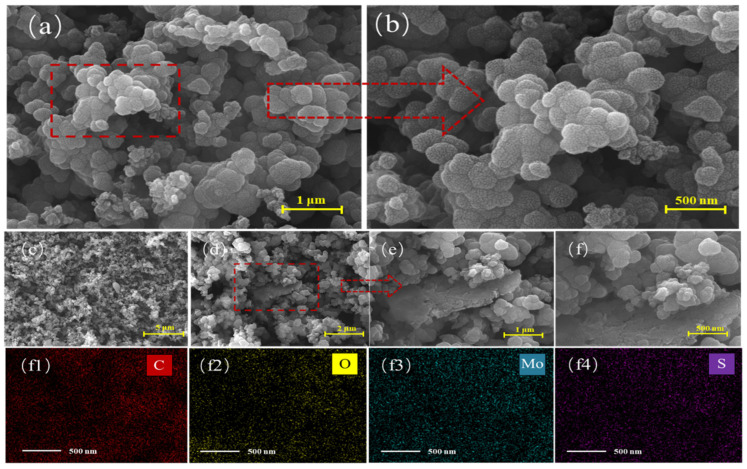
SEM images of (**a**,**b**) the PPy and (**c**–**f**) the PPy/MoS_2_ composite, and (**f1**–**f4**) elemental distribution maps of the view of the composite shown in the SEM image in panel (**f**).

**Figure 4 polymers-16-01940-f004:**
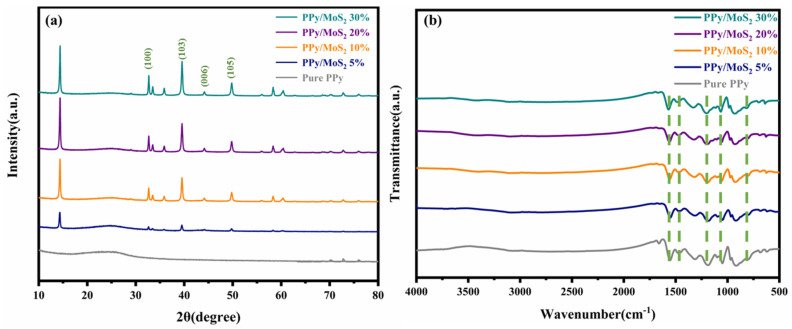
(**a**) XRD patterns and (**b**) FTIR spectra of pure PPy and PPy/MoS_2_ composites with various MoS_2_ mass fractions.

**Figure 5 polymers-16-01940-f005:**
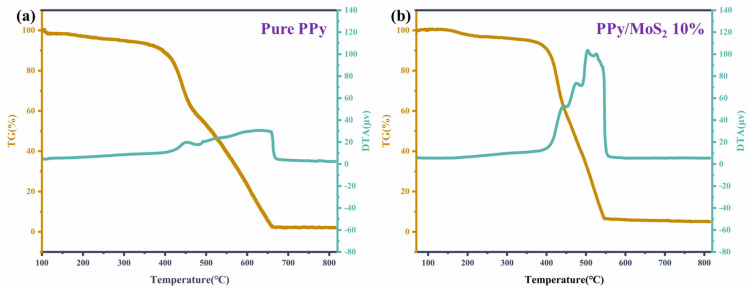
The TG and DTA curves of (**a**) PPy and (**b**) the PPy/MoS_2_ composite.

**Figure 6 polymers-16-01940-f006:**
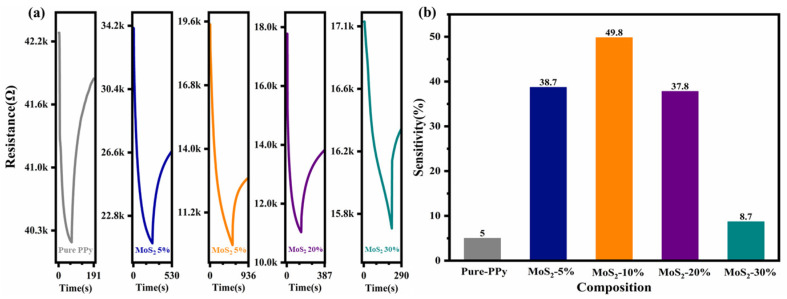
(**a**) Resistance and (**b**) sensitivity of sensors based on composites with different MoS_2_ contents at 100 ppm NO_2_.

**Figure 7 polymers-16-01940-f007:**
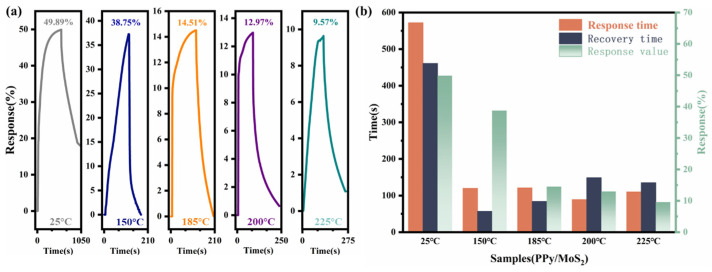
Comparison of (**a**) sensitivity and (**b**) response/recovery times of 10% PPy/MoS_2_ sensor at 100 ppm NO_2_ and different temperatures.

**Figure 8 polymers-16-01940-f008:**
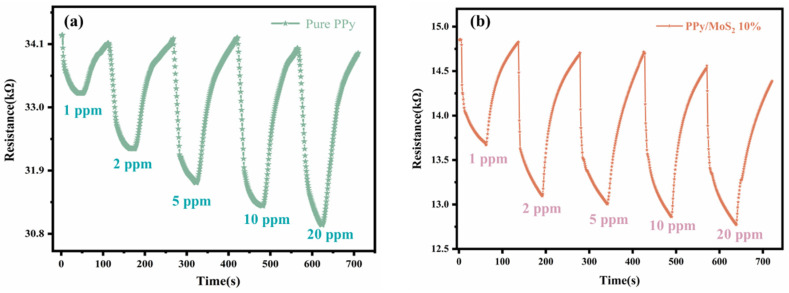
The change in resistance with a NO_2_ concentration in the range of 1 to 20 ppm at 150 °C for the (**a**) pure PPy and (**b**) PPy/MoS_2_ 10% sensors.

**Figure 9 polymers-16-01940-f009:**
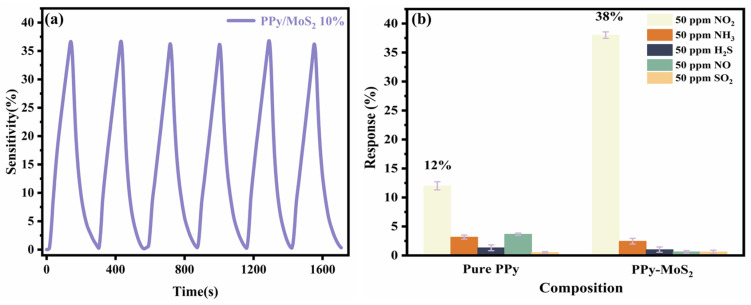
(**a**) Repeatability at 150 °C and 100 ppm NO_2_ of PPy/MoS_2_ sensor and (**b**) selectivities of PPy and PPy/MoS_2_ sensors for NO_2_.

**Figure 10 polymers-16-01940-f010:**
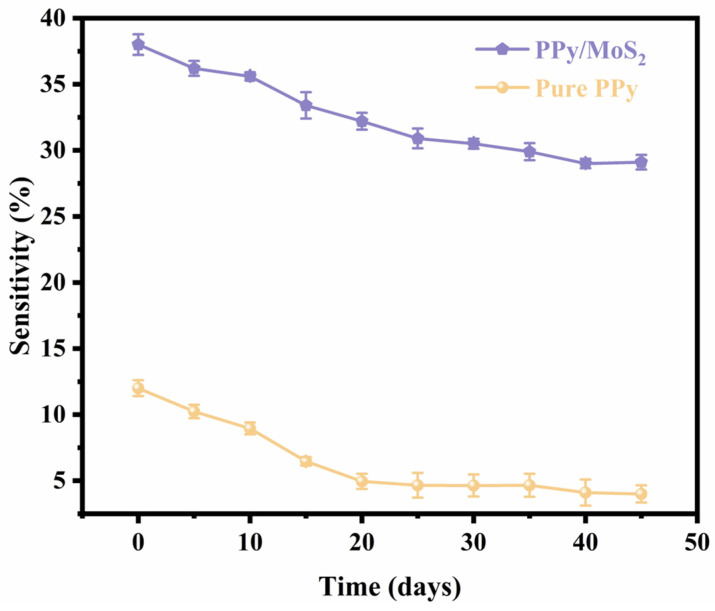
The long-term stabilities of the PPy and PPy/MoS_2_ sensors.

**Figure 11 polymers-16-01940-f011:**
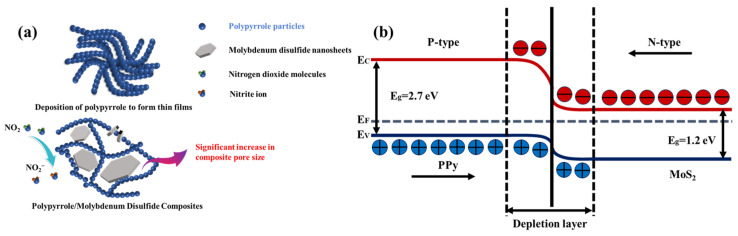
(**a**) A schematic diagram of the mechanism underlying the enhanced performance of the PPy/MoS_2_ composite sensor. (**b**) A schematic energy diagram of the PPy–MoS_2_ heterojunction.

**Table 1 polymers-16-01940-t001:** Performance parameters and operating temperatures of several common types of NO_2_ gas sensors.

Material	Structure	Temperature	Concentration	Response	Response/Recovery Time	Reference
MWCNT-WO_3_	Nanocomposite	RT	100 ppm	96%	10 s/20 min	[38]
Pt/AlGaN/GaN	HEMT	300 °C	900 ppm	33	27 min/-	[39]
MWCNT and rGO-WO_3_	Nanoparticles	RT	5 ppm	17%	7 min/15 min	[40]
PTH	Film	RT	100 ppm	33%	400 s/1600 s	[41]
PPy/MoS_2_	Film	150 °C	100 ppm	48.9%	120 s/57 s	This work

## Data Availability

The original contributions presented in the study are included in the article/Appendix A, further inquiries can be directed to the corresponding author.

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
