# Peer review of "Flexible Resistive Gas Sensor Based on Molybdenum Disulfide-Modified Polypyrrole for Trace NO2 Detection"

_polymers, 2024, doi:10.3390/polym16131940_

Round 1

Reviewer 1 Report

Comments and Suggestions for Authors

Could you provide more details about your experimental setup?
1.What was the volume of your test chamber?
2.What volume of the injection gas was used to achieve 1 ppm of NO2?
3.Did you use a simple laboratory syringe as the gas dispenser? How did you fill the gas into the syringe? Was it from a gas bag or a cylinder?
4.How was the gas desorption achieved? Did you simply open the chamber?
5.What is the influence of humidity on your sensor's performance?
6.Could you provide the layout details of the silver fork electrode? Specifically, what are the electrode width and gap width?
7.A real photo of your sensor would be helpful. Additionally, what are the sensor's dimensions?

Author Response

Responses to Reviewers’ comments

Flexible Resistive Gas Sensor Based on Molybdenum Disul-fide-Modified Polypyrrole for Trace NO2 Detection

Manuscript ID: 3070669

We thank the reviewers for their valuable feedback. Below, we have provided a point-by-point response to the reviewers’ comments.

1.What was the volume of your test chamber?

Thank you for your valuable feedback. We apologize for the oversight because of which the volume of the air chamber was not specified. In this work, the volume of the air chamber was 1 L. This has been added in the revised manuscript.

2.What volume of the injection gas was used to achieve 1 ppm of NO2?

The original gas concentration of NO2 we used was 2000 ppm, and we used a 1 ml needle to extract 0.5 ml of gas for injection.

3.Did you use a simple laboratory syringe as the gas dispenser? How did you fill the gas into the syringe? Was it from a gas bag or a cylinder?

We used a syringe to inject gas into the chamber. First, we transferred a small portion of the raw gas from the gas cylinder to a dark bag, and then used a 1 ml needle to extract 0.5 ml of 2000 ppm concentration NO2 gas and inject it into a 1 L chamber.

4.How was the gas desorption achieved? Did you simply open the chamber?

As you mentioned, in this work, the gas testing platform was placed inside a high-power fume hood. When the gas chamber door was opened, the gas began to desorb from the sensor.

5.What is the influence of humidity on your sensor's performance?

Humidity can indeed affect sensor performance. We have placed the sensitivity–humidity relationship diagram in Supplementary Document 4.

6.Could you provide the layout details of the silver fork electrode? Specifically, what are the electrode width and gap width?

During the preparation process, we used a dispensing method to print the forked finger electrode. The inner diameter of the dispensing needle was 0.16 mm. The width of the interdigital electrode gap was approximately 0.8 mm.

7.A real photo of your sensor would be helpful. Additionally, what are the sensor's dimensions?

We fully agree with your point of view. The size of the sensor is approximately 1.5 cm × 1.5 cm. We have added the physical sensor image in Supplementary Document 1.

Reviewer 2 Report

Comments and Suggestions for Authors

The paper “Flexible Resistive Gas Sensor Based on Molybdenum Disulfide-Modified Polypyrrole for Trace NO2 Detection”  is an interesting scientific study. The authors have designed and studied a promising sensor based on polypyrrole (PPy) doped with MoS2. However, there are a number of recommendations that can significantly improve the article:

  1. It is not clear from the text of the article how many such sensors have been produced and studied by the authors, and there are no error bars on all the graphs. It seems that the prototype of the sensor exists in a single copy. If this is the case, the authors should repeat their research and confirm that the preparation method they developed is reproducible and the new sensors have similar characteristics

  2. In most of the experiments the concentration of 100 ppm NO2 is used, but the authors do not explain why this concentration was chosen. One of the key parameters of this type of sensor is sensitivity, so testing the basic characteristics of the sensor at lower concentrations could improve the results

  3. The authors cite flexibility and resistance to mechanical damage as one of the advantages of the sensor they have developed, but there are no experiments to confirm this. Bending and mechanical pressure can damage the sensor and impair its function. It may be premature to cite flexibility as an advantage

  4. The addition of a table comparing the characteristics of the sensor with those of other analogous devices would enhance the article.

  5. It is evident that there has been a confusion between the references to Figures 3 and 4 within the text (line 207).

  6. The authors test the effects of temperature on sensor degradation and show that MoS2 content affects this process (lines 217-233), but the text does not say how much MoS2 the sensor contains in this experiment. The MoS2 content should also be labelled in Figure 5b.

  7. Figure 5 could be enhanced by aligning both graphs on the same scale, with respect to temperature and DTA parameter.

  8. Figure 6a can be improved by signing the MoS2 content for each curve 

  9. The authors conclude that 150 °C can be considered a good working temperature for the sensor (264-265). However, the authors should clarify how the sensor will maintain this temperature continuously and where it makes sense to use such a sensor 

  10. The authors have clearly shown that temperature affects the main characteristics of the sensor, but different experiments were carried out at different temperatures.  For example, the sensitivity was determined at 150°C, but the repeatability was studied at room temperature. In the selectivity and long-term stability studies, the temperature is not specified at all. The authors should decide at what temperature their sensor will operate in the future and investigate all the characteristics at that temperature.

  11. Figure 8 shows a comparison of the sensitivity of pure PPy and 10% PPy/MoS2 sensors at 150°C. However, 150°C is the optimum temperature for PPy/MoS2 only. It would be interesting to see a graph of pure PPy at its own optimum. For example, this parameter can be compared at 150°C and 20°C for both systems

  12. In figure 9 the authors measure the repeatability. However, 3 simultaneous measurements are not enough, so increasing the number of repetitions will make the results more reliable.

  13. Lines 283-284 say "Over three cycles of measurements at room temperature, the average response was 38%", whereas above (247-248) it says the response is 49% at room temperature. it may be a typo, but it can undermine the credibility of the article.

Author Response

Responses to Reviewers’ comments

Flexible Resistive Gas Sensor Based on Molybdenum Disul-fide-Modified Polypyrrole for Trace NO2 Detection

Manuscript ID: 3070669

We thank the reviewers for their valuable feedback. Below, we have provided a point-by-point response to the reviewers’ comments.

  1. It is not clear from the text of the article how many such sensors have been produced and studied by the authors, and there are no error bars on all the graphs. It seems that the prototype of the sensor exists in a single copy. If this is the case, the authors should repeat their research and confirm that the preparation method they developed is reproducible and the new sensors have similar characteristics

Thank you for your valuable feedback. We fully agree with the issues you pointed out. In fact, we prepared at least 10 sensors for each component and tested them, and the test results show a relatively consistent trend. During the preparation process, the overall dispersion of the material was good, so the prepared sensor has good repeatability. Given the constraints regarding the length of the article, we were unable to provide plots of all the data. At the same time, we believe that your suggestion is highly relevant. We have supplemented the long-term stability and selectivity images of the sensor and added error bars.

  1. In most of the experiments the concentration of 100 ppm NO2 is used, but the authors do not explain why this concentration was chosen. One of the key parameters of this type of sensor is sensitivity, so testing the basic characteristics of the sensor at lower concentrations could improve the results

Testing at lower concentrations can indeed better reflect the performance of the sensor. NO2 gas causes harm to human health at around 20 ppm. We have also tested the response recovery curves of two types of sensors at concentrations of 1–20 ppm, and we fully agree with your point of view. We considered whether organic-inorganic composite materials would fail or become toxic at high concentrations. Accordingly, we chose a concentration of 100 ppm for testing at the beginning of the article and even in most experiments, and selected the optimal composition and temperature. Experimental data show that the material still maintains good impact resistance and good repeatability in a high-concentration gas environment at 5 times the range.(line 236)

  1. The authors cite flexibility and resistance to mechanical damage as one of the advantages of the sensor they have developed, but there are no experiments to confirm this. Bending and mechanical pressure can damage the sensor and impair its function. It may be premature to cite flexibility as an advantage

We fully agree with your viewpoint that the performance of sensors is indeed significantly affected by bending angles. Therefore, we have added a graph of the relationship between sensor sensitivity and bending angles in Supplementary Document 5.

  1. The addition of a table comparing the characteristics of the sensor with those of other analogous devices would enhance the article.

In response to your comment, we have added Table 1 to highlight the performance advantages of the prepared flexible sensor in comparison with other devices.(line 285)

Material

Structure

Temperature

Concentration

Response

Response-recover time

Reference

MWCNT-WO3

Nanocomposite

RT

100 ppm

96%

10 s/20 min

[38]

Pt/AlGaN/GaN

HEMT

300 °C

900 ppm

33

27 min/-

[39]

MWCNT and rGO -WO3

Nanoparticles

RT

5 ppm

17%

7 min/15 min

[40]

PTH

Film

RT

100 ppm

33%

400 s/1600 s

[41]

PPy/MoS2

Film

150 °C

100 ppm

48.9%

120 s/57 s

This work

  1. It is evident that there has been a confusion between the references to Figures 3 and 4 within the text (line 207).

We apologize for the error in the citation. We have made the corresponding corrections in response to your comment.(line 205)

  1. The authors test the effects of temperature on sensor degradation and show that MoS2 content affects this process (lines 217-233), but the text does not say how much MoS2 the sensor contains in this experiment. The MoS2 content should also be labelled in Figure 5b.

There was indeed a lack of clarity. We have made corresponding revisions according to your feedback.(line 217)

  1. Figure 5 could be enhanced by aligning both graphs on the same scale, with respect to temperature and DTA parameter.

We have made proportional modifications to Figure 5 according to your feedback.

  1. Figure 6a can be improved by signing the MoS2 content for each curve 

The quality score of MoS2 was not clearly marked in Figure 6a. We have made the necessary modifications to Figure 6 in accordance with your feedback.

  1. The authors conclude that 150 °C can be considered a good working temperature for the sensor (264-265). However, the authors should clarify how the sensor will maintain this temperature continuously and where it makes sense to use such a sensor 

We set 150 °C as the optimal working temperature. By maintaining a given constant voltage, the sensor is kept at 150 °C. Comparison with other published studies (Table 1 in the revised manuscript, added in response to Comment 4) shows that the working temperature of the sensor in this work is lower than that of traditional metal oxide sensors. The developed sensor also exhibits a better response recovery time compared to conventional pure conductive polymer thin film sensors.

  1. The authors have clearly shown that temperature affects the main characteristics of the sensor, but different experiments were carried out at different temperatures.  For example, the sensitivity was determined at 150°C, but the repeatability was studied at room temperature. In the selectivity and long-term stability studies, the temperature is not specified at all. The authors should decide at what temperature their sensor will operate in the future and investigate all the characteristics at that temperature.

We apologize for not specifying the testing temperature for sensitivity, room temperature, and long-term stability. In this work, we first selected the optimal component of the composite material at room temperature and then selected a more suitable temperature for the optimal component composite material. Taking into account the sensitivity and response recovery time of the sensor, we selected 150 °C as the optimal temperature point for the sensor. We have made relevant revisions in the article and supplemented the image annotations.(line295, 288,290)

  1. Figure 8 shows a comparison of the sensitivity of pure PPy and 10% PPy/MoS2 sensors at 150°C. However, 150°C is the optimum temperature for PPy/MoS2 only. It would be interesting to see a graph of pure PPy at its own optimum. For example, this parameter can be compared at 150°C and 20°C for both systems

We fully agree with your point of view. In Figure 6, it can be observed that in the case of several components at room temperature, the sensitivity of the pure PPy sensor is 5%, while the sensitivity of the 10% component is 49.8%. To further improve the response recovery characteristics of the sensor, the sensitivity of the sensor slightly decreased at 150 °C; nonetheless, the response recovery characteristics significantly improved. Therefore, we set the temperature at 150 °C. Figures 6a and 7a indicate that both pure PPY sensors and binary composite sensors have long recovery times. In our experimental study, it was difficult for the material resistance to recover to the initial level at room temperature; this would take tens of minutes or even several hours, which is not conducive to the application of sensors. We have provided a pure polypyrrole response recovery curve image, which shows that the sensitivity of the sensor is lower than that of the binary composite sensor.

  1. In figure 9 the authors measure the repeatability. However, 3 simultaneous measurements are not enough, so increasing the number of repetitions will make the results more reliable

In accordance with your valuable feedback, we have added the number of repetitions.

  1. Lines 283-284 say "Over three cycles of measurements at room temperature, the average response was 38%", whereas above (247-248) it says the response is 49% at room temperature. it may be a typo, but it can undermine the credibility of the article.

This was indeed a typographical error on our part. We regret that this mistake hindered clarity. We have made modifications to the corresponding sections as per your instructions. Thank you again for the meticulous review of our article.(line 288,290)

Round 2

Reviewer 1 Report

Comments and Suggestions for Authors

The authors have addressed the issues raised, and this manuscript can be accepted in its current form.